# Pattern Recognition Approach and LiDAR for the Analysis and Mapping of Archaeological Looting: Application to an Etruscan Site

Maria Danese [1], Dario Gioia [1], Valentino Vitale [1], Nicodemo Abate [1], Antonio Minervino Amodio [1], Rosa Lasaponara [2] and Nicola Masini [1,*]

1  Institute of Heritage Science, National Research Council (ISPC CNR), I-85050 Tito, Potenza, Italy; maria.danese@cnr.it (M.D.); dario.gioia@cnr.it (D.G.); valentino.vitale@ispc.cnr.it (V.V.); nicodemo.abate@ispc.cnr.it (N.A.); antonio.minervinoamodio@ispc.cnr.it (A.M.A.)
2  Institute of Methodologies for Environmental Analysis (IMAA CNR), I-85050 Tito, Potenza, Italy; rosa.lasaponara@imaa.cnr.it
*  Correspondence: nicola.masini@cnr.it

**Abstract:** Illegal archaeological excavations, generally denoted as looting, is one of the most important damage factors to cultural heritage, as it upsets the human occupation stratigraphy of sites of archaeological interest. Looting identification and monitoring are not an easy task. A consolidated instrument used for the detection of archaeological features in general, and more specifically for the study of looting is remote sensing. Nevertheless, passive optical remote sensing is quite ineffective in dense vegetated areas. For these type of areas, in recent decades, LiDAR data and its derivatives have become an essential tool as they provide fundamental information that can be critical not only for the identification of unknown archaeological remains, but also for monitoring issues. Actually, LiDAR can suitably reveal grave robber devastation, even if, surprisingly, up today LiDAR has been generally unused for the identification of looting phenomenon. Consequently, this paper deals with an approach devised ad hoc for LiDAR data to detect looting. With this aim, some spatial visualization techniques and the geomorphon automatic landform extraction were exploited to enhance and extract features linked to the grave robber devastation. For this paper, the Etruscan site of San Giovenale (Northern Lazio, Italy) was selected as a test area as it is densely vegetated and was deeply plundered throughout the 20th century. Exploiting the LiDAR penetration capability, the prediction ability of the devised approach is highly satisfactory with a high rate of success, varying from 85–95%.

**Keywords:** LiDAR; looting; visualization techniques; pattern recognition; geomorphon; San Giovenale; Etruscan archaeology; Italy

## 1. Introduction

Illegal archaeological excavations, also referred to as looting, are carried out to steal and sell archaeological treasures. Looting practice constitutes a great loss, not only in terms of loss of archaeological treasure and findings, but mostly in terms of knowledge, being that illegal excavations irremediably devastate the cultural and historical contexts of archaeological findings removed from their own locations, stratigraphy, and geographic context [1,2]. The phenomenon is present all over the world, from Southern America [3,4] to Europe [5] and the Middle East [6,7].

The monitoring and quantification of looting phenomenon are necessary steps to reduce and fight these illegal activities. In the last decade, remote sensing tools have been shown to be quite effective for looting detection in different landscapes and environmental contexts. With this aim, different remote sensing technologies can be suitably applied, even if several works are based on the use of passive optical remote sensing [8,9] applied in desert and/or scarcely vegetated areas.

Laser Imaging Detection and Ranging (LiDAR) is widely used in the archaeological field, for the identification of archaeological remains and large settlements under wooded areas [10,11], micro-topography characterization under canopy [12], and reconstruction of lost past landscapes [13].

Consequently, the challenge today is the study and the identification of the most effective methods for LIDAR data interpretation and pattern extraction. Different approaches with these aims are present in the literature. One of the most effective is based on the digital terrain model (DTM) visualization techniques (VTs), and more specifically on the visual interpretation of the different parameters that is possible to extract with VTs [14]. Some authors go further on VTs, by adding to them an analytical approach by using spatial analysis [15]. Other studies use automatic methods based on machine learning [16,17] or some more specific branch of it, such as deep learning [18]. In fact, some examples are based on convolutional neural networks [19,20], on semantic segmentation [21], on artificial neural network and spatial segmentation [22] and on mask-R CNN [23] for archaeological feature detection.

Even if these methods are obtaining great popularity, they are not always easy to use as they need a large amount of training data, computational power and skills for their development [24]. Moreover, all these methods are rarely used for the study of looting, probably due to the morphological nature of bugs, characterized by an irregular shape, that make it more difficult to realize an effective dataset.

Therefore, the aims of this paper are:

- To propose the use of LiDAR to deepen the investigation of the looting phenomenon in the archaeological area of the S. Giovenale (introduced in Section 2.1), partially covered by woods and strongly affected by clandestine excavations, 'specialized' in Etruscan antiquities, particularly in the second half of the 20th century [25];
- To create a multi-scalar approach, ad hoc created for the looting phenomenon, that starts from the use of VTs (presented in Section 2.4 with the related literature) and ends with a classification model based on the Geomorphon concept. The Geomorphon approach (explained in Section 2.5 with the related references) is commonly adopted in geomorphology for the automatic classification of landforms (ACL, [26]) at a landscape scale. The Geomorphon landform/pattern recognition approach has been herein used and tested to classify and map the "small-scale" land-forms related to looting pits. Maps derived by such an approach have been compared with the results of a field survey aimed at the delineation of the topographic features related to the looting phenomena in order to verify the accuracy of the geomorphon-based approach for the detection of looting phenomena at a wider scale.
- To map and to quantify the entity of the looting phenomenon at S. Giovenale, still not investigated geographically and quantitatively. A field check was carried out by a RTK GPS survey, which allowed us to investigate the accuracy of our method based on the semi-automatic extraction of looting-related pits.

## 2. Materials and Methods

### 2.1. Case Study

The investigated area is in the territory of Brera, at about 60 km NW of Rome (Figure 1), and is located over wide gently-dipping surfaces, which roughly correspond with the depositional top of Middle to Upper Pleistocene pyroclastic deposits. This surface is incised by the Vesca River and its tributary. In this territory, archaeological investigations showed evidence of human occupation from the proto-Villanovan culture up to the medieval period, including Etruscan period. The area of major archaeological interest is the Acropolis of San Giovenale settled from proto-Villanovan to Etruscan period (13th–3rd centuries BC), excavated by the Swedish Institute of Classical Studies in Rome from 1956 to 1965 [27–29]. At East of San Giovenale there is Vignale, another area of archaeologist interest investigated from 2009 to 2011 integrating satellite remote sensing and LiDAR, which revealed the

presence of tombs from the Villanovan period and a road connecting Vignale with the Acropolis [30].

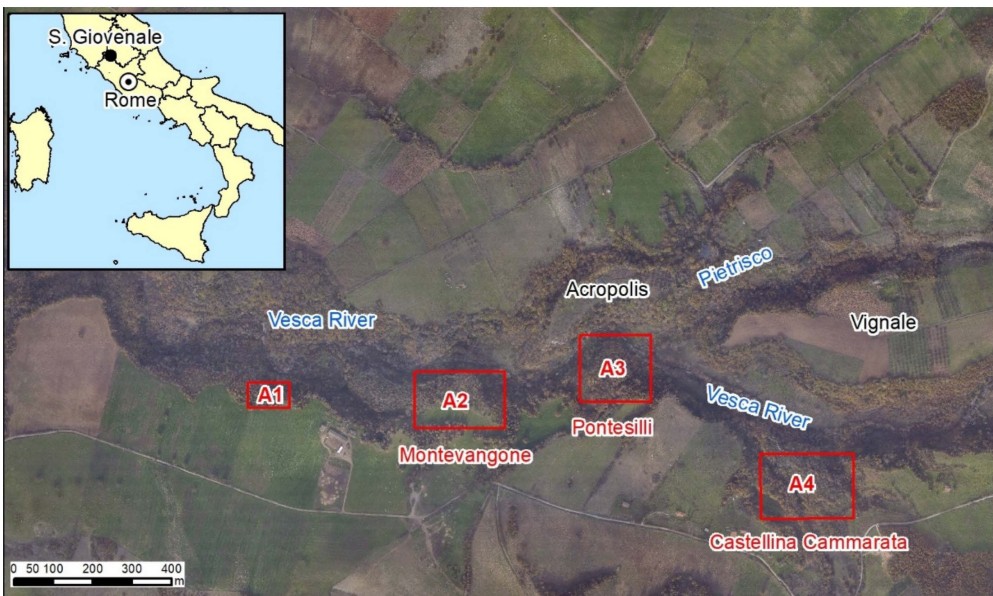

**Figure 1.** The S. Giovenale study area, with the four subset areas and the main placenames.

A less investigated area by archaeologists is at South of San Giovenale and Vignale, and the Vesca river. The dense wood and the clandestine excavation activity probably discouraged research and archaeological excavation projects. In this area, which includes Montevangone, Pontesilli and Castellina Cammarata (see Figure 1), are located the four test areas (indicated as A1, A2, A3, A4 in Section 3) selected for this study.

### 2.2. Methodological Approach

The methodology is composed of the following steps (see Figure 2): (1) LiDAR survey and the data acquisition; (2) LiDAR data processing aimed at generating the DEM; (3) the enhancement by means of the creation of derived models based on VTs to facilitate the identification of looting feature; (4) the pattern-recognition based on Geomorphon, to semi-automatically extract potential looting features; (5) validation of results, through (a) the observation of the looting features from the derived models based on VTs and the topographic profiles on the region of interest and (b) field control of several looting tombs using a GNSS in RTK mode (Trimble R2 receiver and Trimble TSC5 controller). GPS survey has been focused on the detailed delineation of looting-related pits, which has been used as basic data to estimate the general prediction ability of the semi-automatic method for the extraction of looting pits.

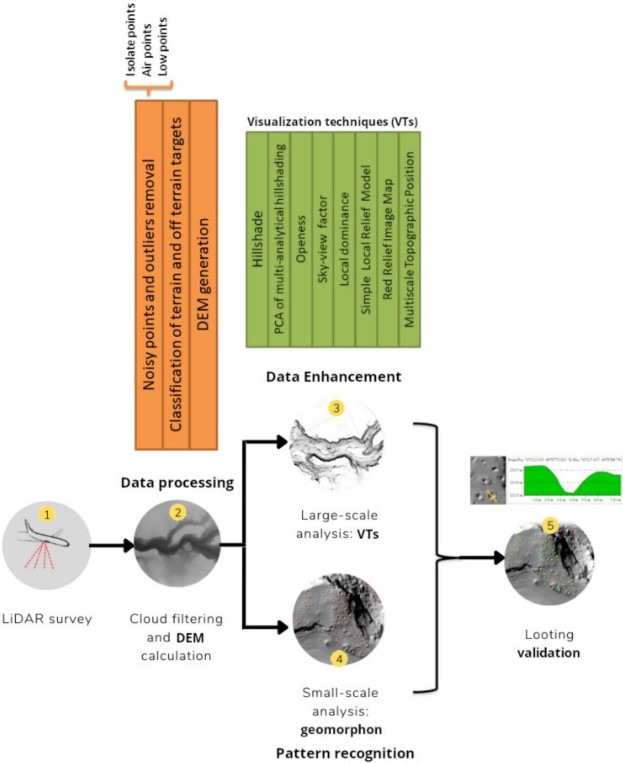

**Figure 2.** Flow chart summarizing the methodology followed to extract looting in S. Giovenale study area.

### 2.3. LiDAR Data Acquisition and Processing

LiDAR survey was carried out on September 2010, over an area of around 6 km$^2$ by using a Full waveform scanner RIEGL LMS-Q560, which digitize the complete waveform of each backscattered pulse, on board a helicopter. The scanner acquired data in South-North and East-West directions, with a divergence of the radius 0.5 mrad and a pulse repetition rate at 180,000 Hz. The average point density value of all the datasets is about 20 points/m$^2$. The accuracy is 25 cm in xy and 10 cm in z (altitude).

The first step of LiDAR data processing consisted in removing noisy points and outliers, including:

- isolated points (when no other cloud points are present in their neighborhoods);
- air points (such as low altitude planes, birds or those points that are far higher from the nearby rough surface);
- and, finally, low points, that are the points lower than their adjacent ground LiDAR points (in the case of San Giovenale, the removal of low points have been set assuming a height lower than 0.5 m, with respect to other points within a ray of 5 m).

The second step has been the classification of terrain and off terrain targets that are crucial for the identification and interpretation of the microtopographical features with truncated shape related to looting, especially in the presence of shrubs and trees, as in the case of San Giovenale, as the reflection generated by low vegetation canopies are often mistaken for those of bare terrains [31].

The DTM (Figure 3) was obtained using the progressive Triangulation Irregular Network (TIN) densification method by Axelsson [32], embedded in Terrasolid's Terrascan [http://www.terrasolid.fi/en/products/terrascan (accessed on 1 March 2022)].

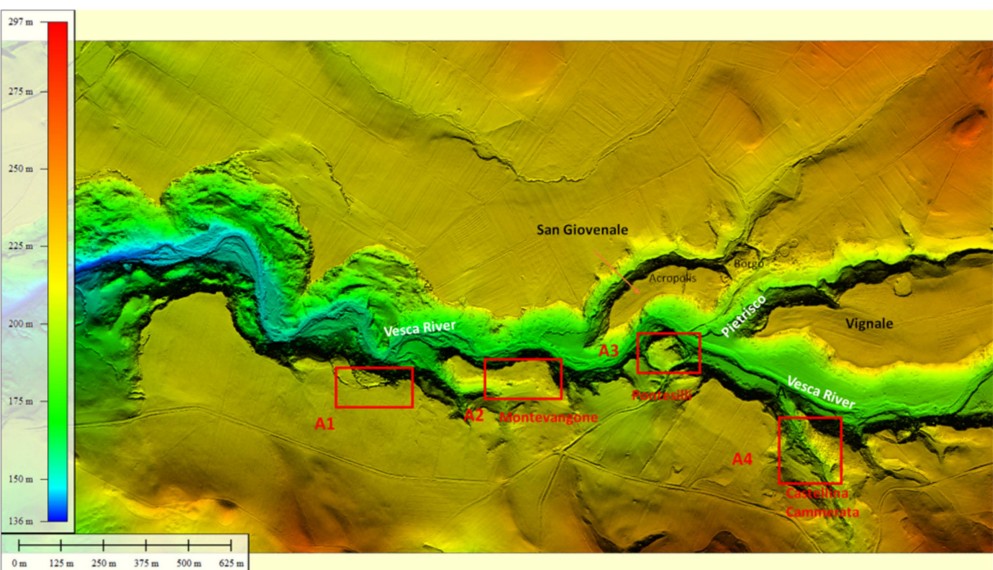

**Figure 3.** The S. Giovenale LiDAR-extracted DEM (above) and the land cover (below), with the indication of the four areas more found to be more affected by looting.

*2.4. Enhancement*

The generated DEM has been post-processed using some visualization techniques, in order to emphasize and facilitate the identification and interpretation of micro topographical features of archaeological interest including looting pits, characterized by tendentially inverted truncated cone shape.

Below, the methods used in this work are briefly presented. Empirical comparison between these methods can be found in the literature [19,33–36].

### 2.4.1. Hillshade (HS)

HS belongs to the first analytical methods used in the literature over DEM derived from LiDAR [37]. In the HS technique, shadows created by the terrain morphology are calculated [33]. To do this, a light source is imposed over the study area, by defining its azimuth and altitude; these parameters strongly influence HS. This is a limitation for the visual interpretation of the resulting HS, as there are some elements that are highlighted by some positions of the light source, and others are hidden [38].

### 2.4.2. PCA of Multi-Analytical Hillshading (MAH)

Due to the limitations of Hillshade, a multi-analytical version of HS was introduced in the literature [39], in order to highlight all the hidden elements. In this method, different HSs are calculated, each by using different light source directions. To combine the resulting HSs, different methods exist, and between these, one of the most common, used in this paper, is Principal Component Analysis (PCA) [40].

### 2.4.3. Openness

Openness is an angular measure of topographic visibility of a territory. In particular, it expresses the dominance of enclosure of all the pixel in one raster [40–42], calculated from fixed points of view, having different azimuth and nadir characteristics. The number and the distance of the points of the view from the viewed pixel (search radius) influences the result of the analysis [36].

Two types of topographic openness can be calculated: positive openness (PO) considers a viewer perspective above the DEM surface, negative openness (NO) under the DEM surface. According to these two perspectives, high PO values highlight convex forms, while high NO values highlight concave forms.

### 2.4.4. Sky View Factor (SVF)

The SVF measures, starting from a DEM, the portion of the visible sky visible from each pixel of the study area [43]. Two parameters are needed for SVF calculation: as in Hillshade, a fixed light source, and as in openness, a distance from the light source to the target pixel (search radius). The resulting raster has values varying from zero, when the sky visibility is completely obscured, to one, when the sky visibility is completely open [43].

### 2.4.5. Local Dominance (LD)

LD is used to evaluate convexity or concavity in the morphology of a territory, through the concept of pixel dominance when the pixel is chosen as an observer position that looks at the territory within a search radius. High values of LD show a huge angle of visibility from the pixel and are suitable for peaks, while low values characterize depressions [44].

### 2.4.6. Simple Local Relief Model (SLRM)

SLRM is a method particularly useful to visually highlight small features from a LiDAR-derived DEM. In fact the aim of SLRM is to minimize large-scale landscape forms in order to leave small-scale features. To do this SLRM derives from the subtraction of the original DEM from the same filtered and consequently smoothed DEM [14]. The filter radius is the most important parameter in this method.

### 2.4.7. Red Relief Image Map (RRIM)

RRIM is a visualization method that combines the slope of the study area and the *PO* and *NO* parameters [44,45]. Firstly *PO* and *NO* are combined as follows (Equation (1)):

$$I = \frac{PO - NO}{2} \tag{1}$$

Then slope and *I* are visually overlaid by using transparencies and red colors, in order to highlight morphological features: the slope is colored with an increasing red gradient; with *I* it is possible to color the ridges in white and valleys in black.

### 2.4.8. Multiscale Topographic Position (MTP)

MTP is a method of automatic landform classification born to go over limits associated to the classical Topographic Position Index (TPI) calculation [46]. In fact, the use of multiple moving windows overcame the high degree of scale dependence of the TPI method [47]. With this aim, MTP integrates hierarchically large, medium and small-scale TPI [47], that can be combined in RGB colors.

### 2.5. Geomorphon Automatic Classification

The geomorphon method is an algorithm of automated landform classification reintroduced by Jasiewicz et al. (2013) [48], which has been widely used as an effective tool to automatically extract geomorphological features at a catchment or sub-regional scales (see for example [26,49–51]). Geomorphon-based automatic classification uses a computer vision approach and evaluates the local surface using the line-of-sight principle in a self-adapting circular windows that scan the DEM. The parameters of the algorithm are the inner and outer search radius, and a flatness threshold. In particular, the outer or maximum search radius (lookup distance) sets the maximum distance for line-of-sight (LOS) calculations for each pixel, which is strictly related to scale recognition of the basic landform class. Such an approach and in particular the use of a self-adapting neighborhood statistics favoured the identification of landforms of different sizes and scales, thus limiting the well-known issue of the scale-dependance of others algorithms of automatic lanform classification such as the TPI-based methods [52,53]. The Geomorphon method returns ten different landform classes: flat, summit, ridge, shoulder, spur, slope, hollow, footslope, valley, and depression. Automatic detection of looting phenomenon can be achieved by

considering its peculiar topographic feature, which is represented by the class "depression" in the geomorphon map.

## 3. Results

The process followed to analyze the San Giovenale area is summarized in the flowchart in Figure 2.

After the LiDAR survey and the LiDAR cloud filtering of vegetation, a DEM with a pixel of 0.5 m was derived. This was analyzed with VTs, in order to visually emphasize which are regions more affected by looting. In Table 1 the parameters chosen for each VS are showed. The 10 pixel (px) parameter is justified by the dimension of bugs, that have the maximum radius nearly 5 m. The parameter choice has been conducted as follows:

- the search radius assumed for PO, NO, SVF, SLRM and SMTP is equal to 10 pixel (px), as the dimension of bugs, that have the maximum radius equal to nearly 5 m;
- The search radial for LD needs a dimension lower and bigger than the bugs diameters to determine the dominance areas. Consequently, it has been chosen by using values found in the literature for features similar in dimension [18];
- All the other parameters are parameters consolidated in the already cited literature in Section 2.4 for each parameter.

**Table 1.** Parameters used for each VT and indication of the software used for the calculation.

| VT | Parameters | Software |
|---|---|---|
| HS | Sun azimuth: 315°; Sun elevation angle: 45° | QGIS |
| PCA of MAH | Sun azimuths: 16 directions; Sun elevation angle: 45°; Number of principal components: 3 | SAGA library for QGIS |
| PO | Number of search directions: 16; Search radius: 10 px | SAGA library for QGIS |
| NO | Number of search directions: 16; Search radius: 10 px | SAGA library for QGIS |
| SVF | Number of search directions: 16; Search radius: 10 px | SAGA library for QGIS |
| LD | Minimum radius: 2 px; Maximum radius: 25 px | Relief Visualization Toolbox (RVT) [53] |
| SLRM | Radius: 10 px | QGIS |
| RRIM | Blending: overlay of slope (70%) and I 30% | QGIS |
| MSTP | Micro scale (Blue): 2 to 10 px; Meso scale (Green): 12 to 1000 px; Macro scale (Red): 1200 to 2000 px | SAGA library for QGIS |

Between the different VTs applied, the best results in terms of looting visibility are obtained by PO, NO, SVF, RRIM. This is due to the dimension, shape and morphological characteristics of pits: small, very regular and well morphologically defined.

Furthermore, HS allows us to highlight looting. SLRM, MSTP and LD probably would be more suitable to extract features with bounding more "smoothed", while LD and SLRM are more effective for features having a minor morphological contrast with the surrounding landscape.

VTs helped to find four main areas with a high concentration of pits (Figures 4–7). All these areas appear to be located over a terrace. This is plausible as this type of landform is a typical positive factor for the settlement choice in the past. In fact, the areas already investigated in other works in the literature [29,54] are also characterized by the presence of the castle and by attested human frequentation are over such a type of morphological element.

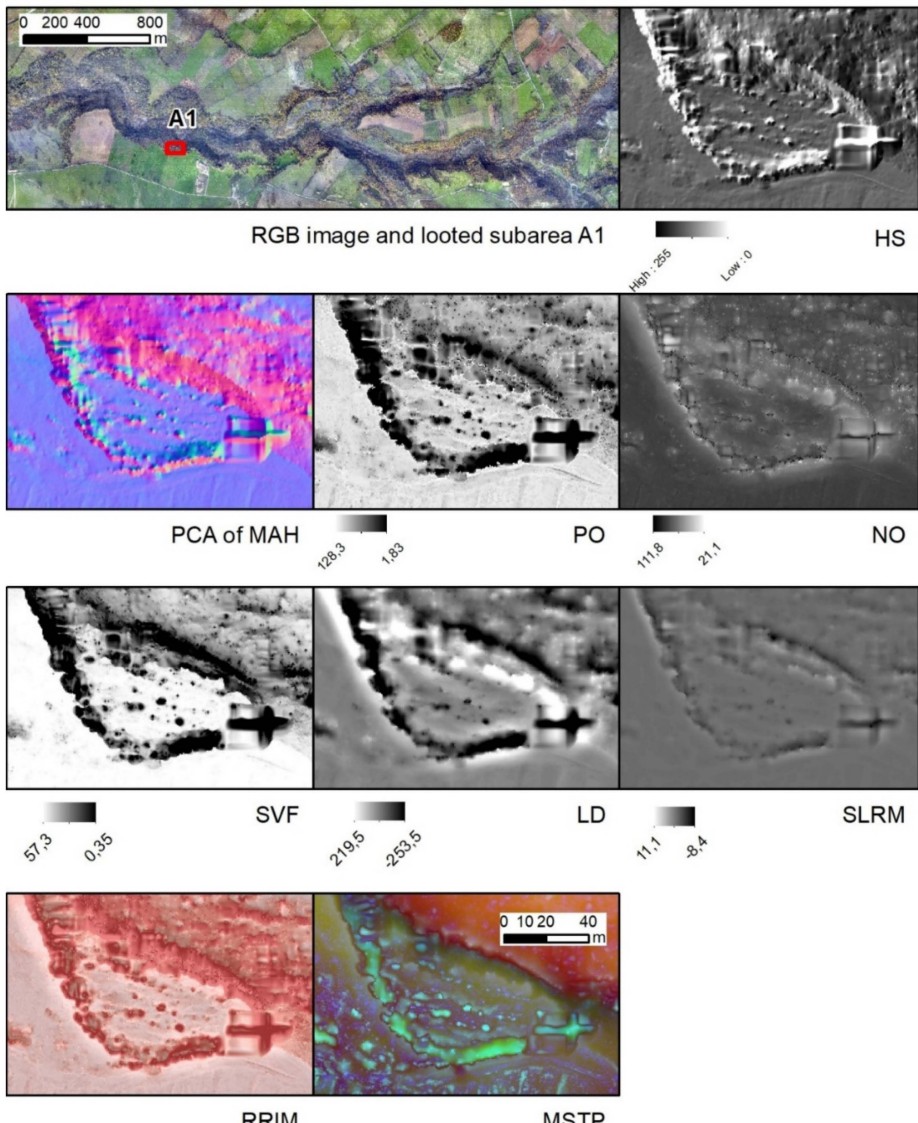

**Figure 4.** The first of the four areas found with VTs affected by looting (A1). In the left-high corner, the localization of the A1 subarea is overlaid to the RGB image. In the other frames the different VT results are showed.

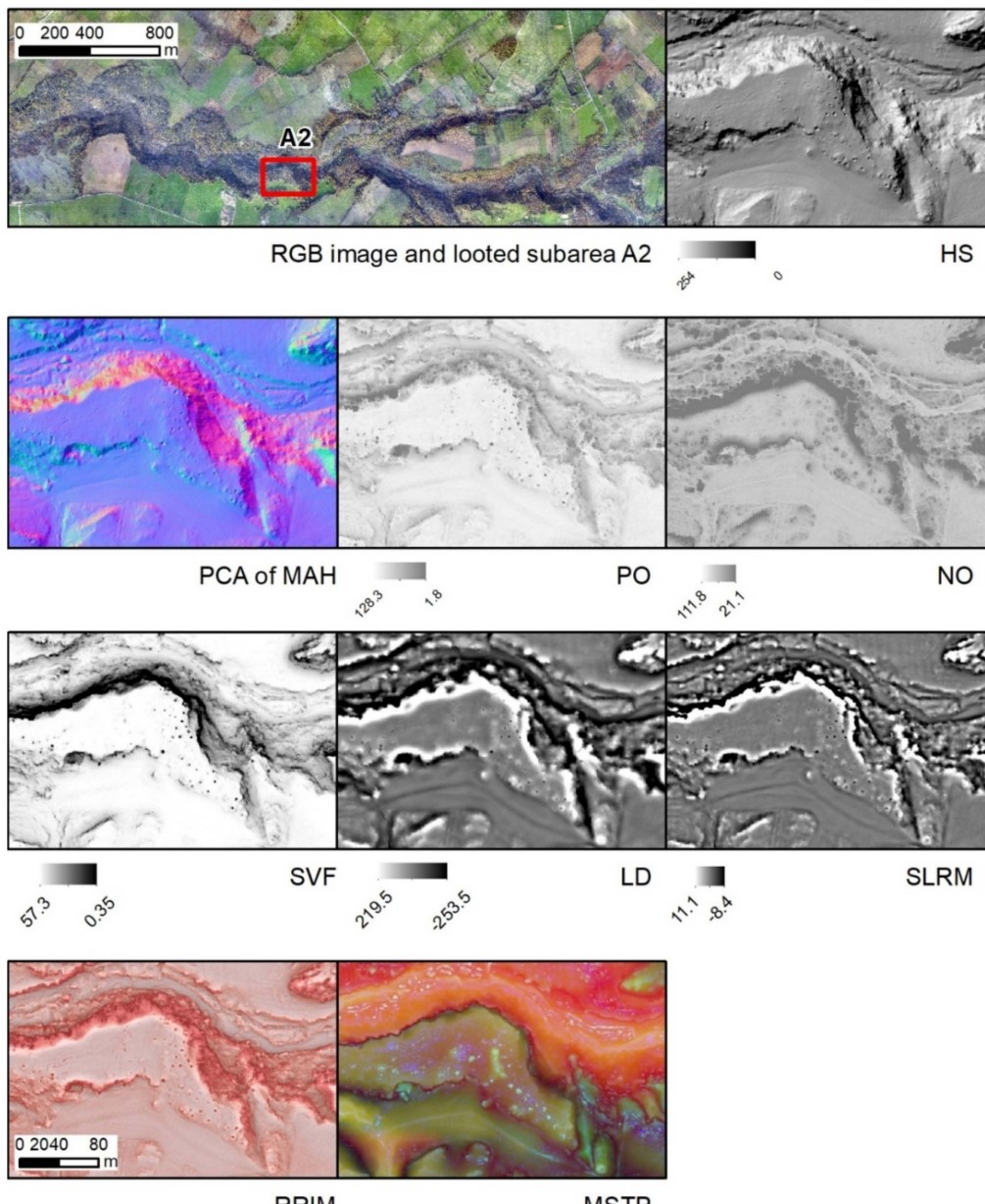

**Figure 5.** The subarea A2 found with VTs affected by looting. In the left-high corner, the localization of the A2 subarea is overlaid to the RGB image. In the other frames the different VT results are showed.

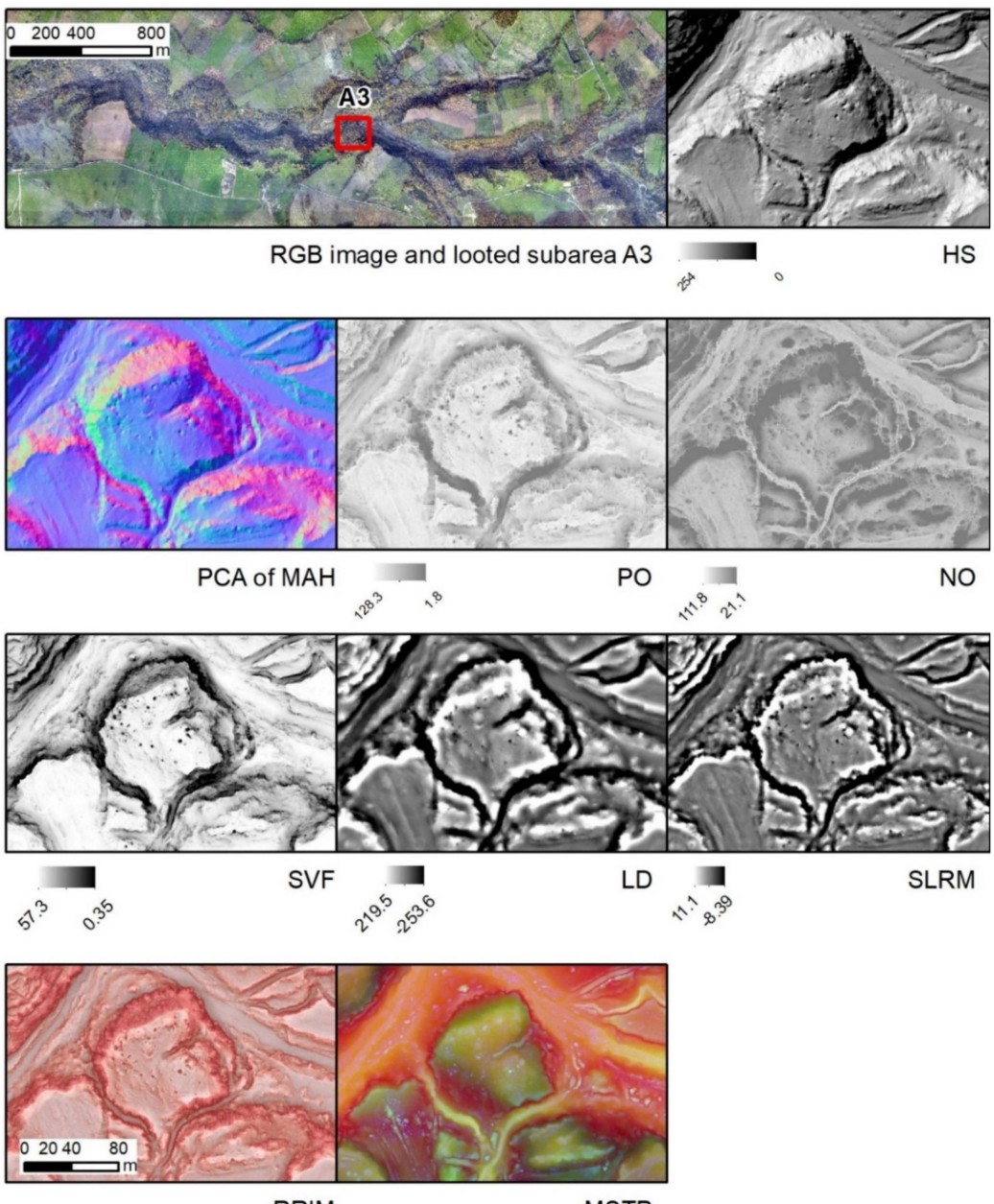

**Figure 6.** The subarea A3 found with VTs affected by looting. In the left-high corner, the localization of the A3 subarea is overlaid to the RGB image. In the other frames the different VT results are showed.

Over these four areas, the Geomorphon method was applied in order to extract and quantify looted areas.

In order to calibrate the geomorphon map to the size of the loooting-related pits, we use an approach consisting of an iterative extraction of geomorphon-based maps with an increasing size of inner and outer search radius, always considering the variations in pit dimensions.

The following final parameters were used: 1° as threshold angle for flat areas detection, 4 m as outer search radius and 1 m as inner search radius. The best result was evaluated by comparing visually looted areas with the Geomorphon result. Looting corresponds to the landform associated from the model to the code 10 and interpreted as depression.

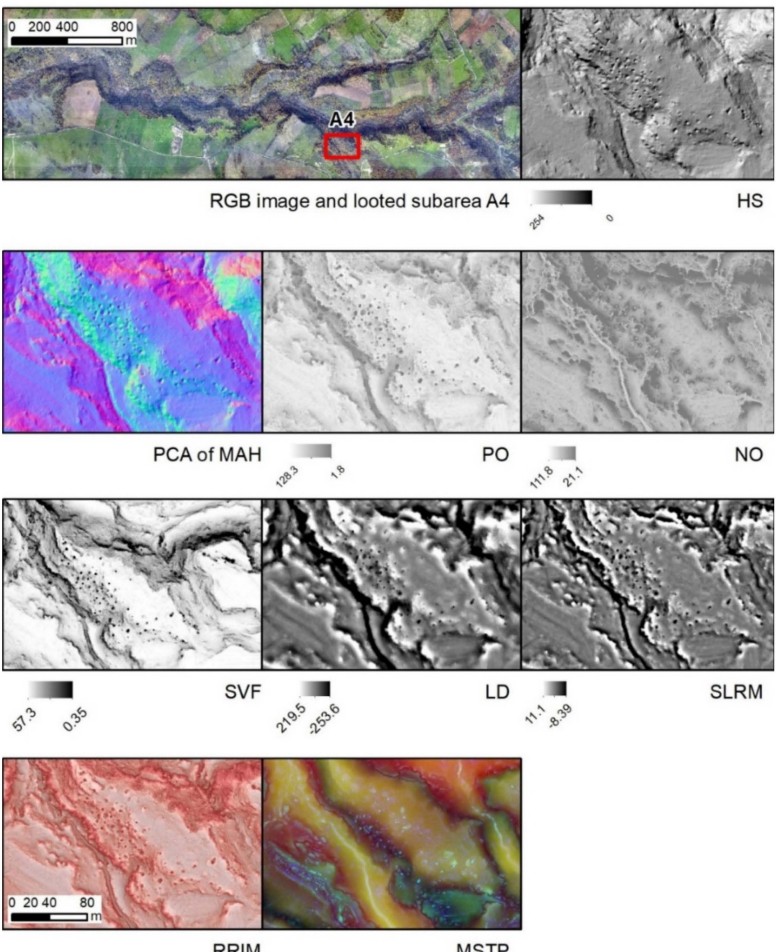

**Figure 7.** The subarea A4 found with VTs affected by looting. In the left-high corner, the localization of the A4 subarea is overlaid to the RGB image. In the other frames the different VT results are showed.

A preliminary visual inspection of the geomorphon-based map shows its viable reliability in detecting looting-related topographic pits. However different filter steps are needed to produce a more effective result (Figure 8):

1. areas lower than 0.25 m² are excluded, as they usually do not correspond to pits;
2. areas higher than 4 m² are deleted too, as they reveal a different type of morphological feature, not related to human excavation practice;
3. with zonal statistics, the mean slope of features was calculated and classified in five quantiles. Considering the particular morphology of pits, features having a mean slope falling in the first of these five quantiles represents topographic features related to geomorphological processes, or, in rare cases, old pits nowadays eroded and quite disappeared;
4. from the VTs it was possible to individualize four areas, characterized by a well-delimited morphology and by a huge density of pits. These four areas were drawn with visual inspection and used to mask the final pits.

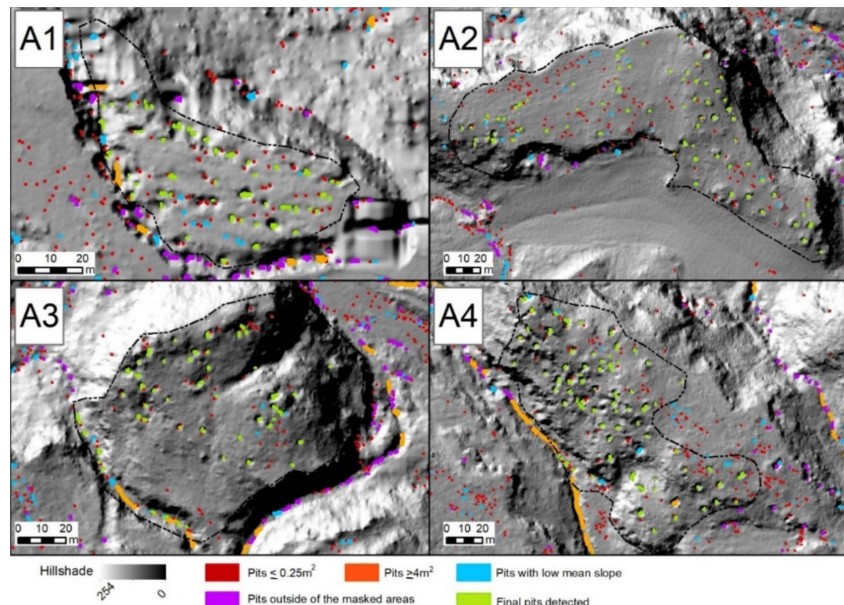

**Figure 8.** Geomorphon results on the four areas of interest. The VT map herein showed is HS.

From the second over imposed filter, less than the areas of the biggest pits, an interesting consideration about the effectiveness of the Geomorphon method is derived: the method is effective to find where looted areas are, and to drawing them, yet only the internal most concave part of pit extracted as valley. To extract all the excavated areas, other landform should be considered, however, this would produce more noise in the results.

For what concerns the assessment of the rate of success of Looting pit detection, the following was conducted:

- exploiting the enhancement of VTs
- controlling the profile of each extracted element in order to control its morphology, as illustrated in Figure 9, along the main pit axes.

The Geomorphon method is able to draw the boundaries of looting, however some elements are not detected (Table 2, Figure 10); moreover, there is also a part of the elements that represent natural little depressions in the terrain that are difficult to be separated automatically from looting and that were excluded by applying a visually overimposed mask, based on a visual combination of morphology and pit density.

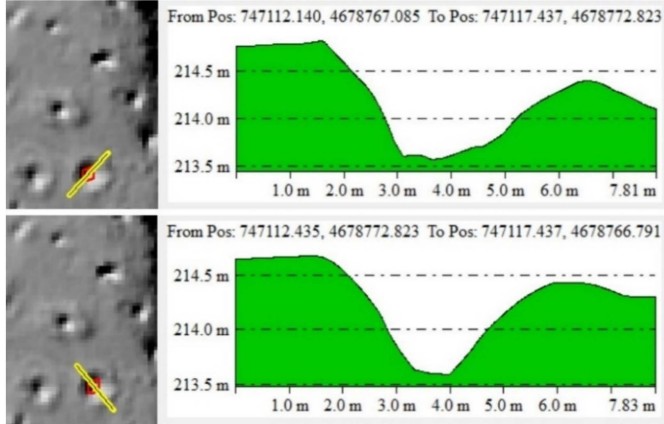

**Figure 9.** *Cont.*

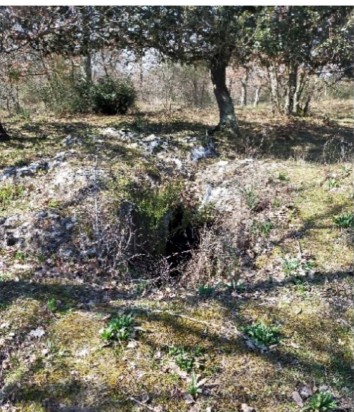

**Figure 9.** Validation conducted exploiting the enhancement of VTS (on the left), the topographic profile from the DEM (on the right) and ground check. The VT herein showed is HS. Example of profiles derivates for a pit extracted (red), along its main axes (yellow).

**Table 2.** Validation of pits: areas occupied by pits extracted correctly (checked), false positives and pits not detected.

| Area | Number of Pits Checked | Number of Pits Not Detected | % of Pits Correctly Extracted |
|------|------------------------|------------------------------|-------------------------------|
| 1 | 52 | 9 | 85 |
| 2 | 72 | 6 | 92 |
| 3 | 54 | 3 | 95 |
| 4 | 83 | 11 | 88 |

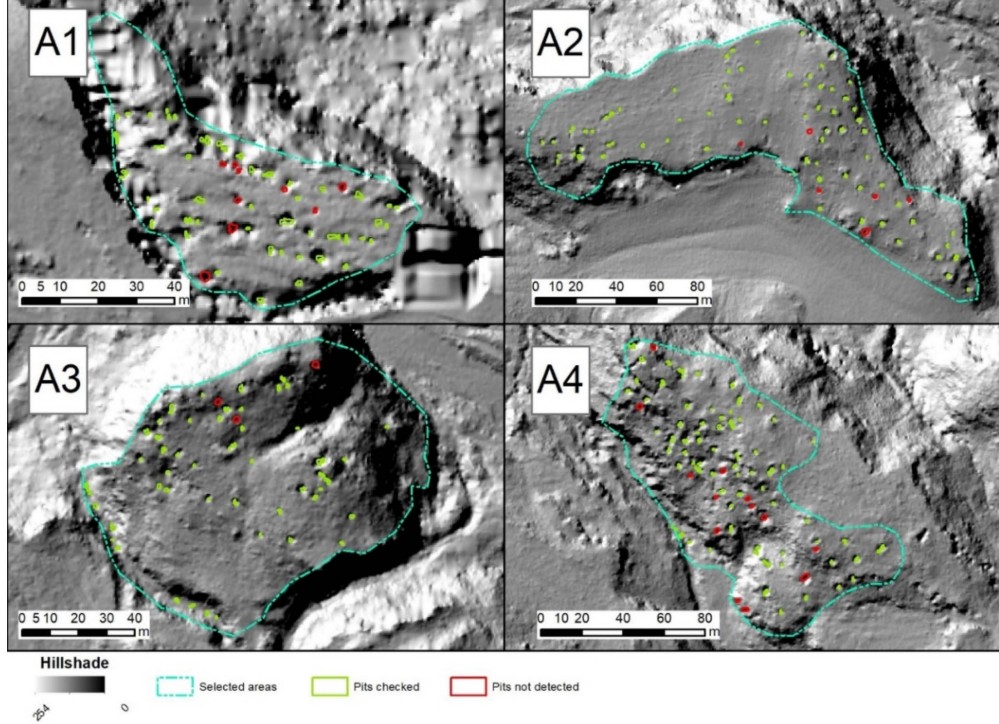

**Figure 10.** Validation of results. Pits extracted with Geomorphon and validated (checked, in green) and pits not detected (in red) for each of the four selected areas.

## 4. Discussion

In this paper we demonstrated the utility of the Geomorphon method to extract looting features from LiDAR-derived DTM. Looting-related pits are small-scale elements with a

peculiar topographic signature. In fact, they are featured by: (i) a size ranging from 0.5 m$^2$ to 7 m$^2$; (ii) a circular or sub-circular shape; (iii) a non-random spatial distribution.

After the visual analysis conducted with the help of different VTs (HS, PCA of MAH, PO, NO, SVF, LD, SLRM, RRIM, MSTP) it was possible to observe that, among them, the more suitable shape characterizing the features are PO, NO, SVF, and RRIM.

In fact, they allowed the selection of four areas largely affected by looting, but only qualitatively from the resulting raster. These areas show similar features from the morphological point of view, as they fall over terraces and near the main stream of the study area. These landscape sectors are areas of preferential occupation of ancient settlements and that could reveal the presence of undiscovered human traces. This would require further investigation and most of all, a field survey. Anyway, our approach, based on the integration of VTs and geomorphon map, is able to individuate four different sectors of the study area with a high degree of small-scale depression, which can be largely ascribed to looting phenomena. Such features cover a total area of 175 m$^2$, which can be mainly attributed to abusively excavated surfaces. Moreover, there are other limitations in the present study. First, our approach only allowed extraction from the internal part of the pits, and consequently, the individualized value is rounded down. In addition, some pits are completely not detected, although most of all, the method is not able to differentiate natural pits present in the landscape, so further study could deepen this aspect. In this case, however, it is very clear which pits were subjected to looting, and which were not, thanks to the visible terraces. Consequently, the methodology proposed in this work could be defined as semi-automatic, as supervision is required to clean the final results.

## 5. Conclusions

For bare areas, optical remote sensing and SAR offer suitable tools for the detection of looting, whereas in wooded areas LiDAR is the unique technology that can provide detailed information under canopy, including archaeological disturbance. The results herein obtained evidence of the high potentiality of LiDAR to detect and map a looted area at a single 'pit', even if completely covered by trees as evident from field survey.

Given the considerable extension of wooded areas in Italy of potential archaeological interest, it is necessary to identify them with appropriate tools such as the LiDAR to understand if they have been disturbed by grave robbers in order to strengthen on-site surveillance and safeguard measures.

Considering the different papers present in the literature and briefly presented in the introduction, and the results obtained in this paper, we conclude that there is the need for methods more effective for pattern recognition and quantification in the study of the looting phenomenon, although that could be an "instant" instrument, easy and fast to perform in order to be an effective instrument for archaeological reasoning and evaluation.

The Geomorphon method could offer a first approach useful to this aim, as it is able to extract the morphological shape that characterize looting. In fact, usually, works already cited in the introduction and in the methods section, which referred to VTs, only performed an enhancement of archaeological patterns. Instead, with Geomorphon, we obtain a percentage of success (here varying from 85 to 95% in the four areas) comparable with the papers that use machine or deep learning. In fact, there, the detection rate varies from 89.5% [17] to above 90% [22] even if, as already underlined, the direct comparison is difficult as the extraction methods are applied on other types of archaeological features, not on looting.

However, another output obtained in this paper shows that further analyses and methods are needed to be deepen as the use of LiDAR with drones in order to improve the: (i) temporal; and (ii) spatial resolution of image acquisition, thus also facilitating the discrimination between looting and 'natural' pits.

In fact, the Geomorphon method allows a semi-automatic approach that still needs human intervention to discriminate between some natural pits that have a morphology similar to clandestine excavation.

The result of field surveys allowed us to verify that the features found by the LiDAR data and extracted using Geomorphon actually referred to clandestine excavations

Most of them are practiced with a localized excavation close to the opening of the tomb chamber, immediately after the *dromos* (entrance passage). The latter are almost never affected as the expected artifacts are in the tomb and not along the dromos (Figure 11).

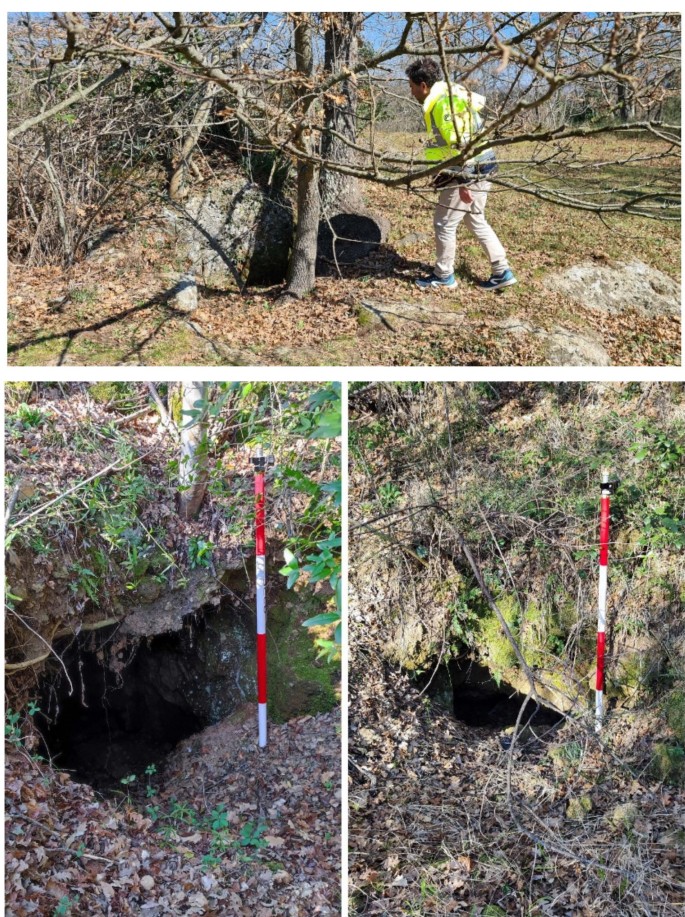

**Figure 11.** Field survey. **Above**: evidence of looting pit close and under a tree; **below**: details on the opening to two tomb chambers desecrated by grave robbers.

In conclusion, the following future research directions are:

- A first aim should be to replicate the approach presented in this paper in other archaeological locations, in order to better validate it and its effectiveness;
- It would be interesting to conduct a direct and empirical comparison between machine learning methods applied to looting extraction, and the analysis with VTs plus Geomorphon, as this is completely missing in the literature;
- Finally, more efforts could be made regarding the combination of Geomorphon with machine learning, to improve the automation of the process of pattern recognition.

**Author Contributions:** Conceptualization, M.D. and N.M.; methodology, M.D., D.G., N.M. and R.L.; software, M.D.; validation, M.D., D.G., N.M. and R.L., A.M.A., V.V.; data curation, M.D., D.G., N.M. and R.L.; writing—original draft preparation, M.D., D.G., N.M. and R.L.; writing—review and editing, M.D., D.G., N.M., R.L, A.M.A., V.V. and N.A. All authors have read and agreed to the published version of the manuscript.

**Funding:** This research received no external funding.

**Institutional Review Board Statement:** Not applicable.

**Informed Consent Statement:** Not applicable.

**Data Availability Statement:** Not applicable.

**Conflicts of Interest:** The authors declare no conflict of interest.

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
