# Peer review of "Pattern Recognition Approach and LiDAR for the Analysis and Mapping of Archaeological Looting: Application to an Etruscan Site"

_remotesensing, doi:10.3390/rs14071587_

Round 1

Reviewer 1 Report

The subject which consists in studying the application of Lidar for the detection of looting is interesting and thus constitutes a corollary to the numerous applications of Lidar for the identification of archaeological traces. Lidar has been used to study historical excavation areas (mines, quarries, excavation areas, trenches, bombs impacts, etc.); this is even more the case for modern or even very recent looting.

Beyond the chosen case study (despite the paper proposes 4 small areas, they constitute a single similar context, rather similarly chosen for the same characters), the paper first declines at length the various - classic - visualization and enhancement tools of a standard GIS toolbox. They are +/- developed here (100 lines and 40 images in 4 figures), even if they confirm the usual - and well known - use of Hillshade... It is on this basis that an application of a Geomorphon-based automatic classification is compared and evaluated. A quite positive evaluation, without much surprise, given its operating principles and capacity to return ‘depression’ class.

However, one may wonder about the scope and relevance of the exercise presented in this article, given the empirical nature of the method, the preponderance of the choice of parameters used, and the centrality of the visual evaluation. Further, the article and the method suffer from a disappointing shortcoming: the absence of any field confrontation, or real reconnaissance or verification on the ground. Therefore, reaching the famous but over-used conclusion of "further analysis and methods are needed to be deepened" (l.372) also suggests that this article is perhaps incomplete and premature. Such field checking – or at least a test zone - doesn’t seems to have involved such a paramount task as the features at stakes only « cover a total area of 175m2 »… In particular, when, in the course of a sentence (l.352), one learns that the method does not allow to distinguish the sought-after features (looting) from natural depressions which could also exist?

Additional remarks:

Somehow, introduction is an exercise where authors show their knowledge of the state of the art and mobilize references, inflating the overall relevance of their contribution. Here, they are plentiful: more than 10 references just to mention global archaeological looting, as many to talk about Lidar, VTs, automated methods etc. One gets 37 references in less than 80 short lines. But not a single one dealing with lidar use of ancient diggings or excavations. See for instance https://journals.openedition.org/echogeo/14791

Basically, since the paper deals with tangible features in specific spaces, it is worrisome that the maps and figures almost never have a scale! Or when they do – rarely -, they are odd (figure 3). Furthermore, consistency in the colour codes used between figures would be preferable (e.g., Figures 8 and 9). One notes in passing the absence of a legend for the blue dots in figure 9(A4).

Author Response

Authors: We thank the reviewer for the valuable comments which enabled us to improve the paper

The subject which consists in studying the application of Lidar for the detection of looting is interesting and thus constitutes a corollary to the numerous applications of Lidar for the identification of archaeological traces. Lidar has been used to study historical excavation areas (mines, quarries, excavation areas, trenches, bombs impacts, etc.); this is even more the case for modern or even very recent looting.

Beyond the chosen case study (despite the paper proposes 4 small areas, they constitute a single similar context, rather similarly chosen for the same characters), the paper first declines at length the various - classic - visualization and enhancement tools of a standard GIS toolbox. They are +/- developed here (100 lines and 40 images in 4 figures), even if they confirm the usual - and well known - use of Hillshade... It is on this basis that an application of a Geomorphon-based automatic classification is compared and evaluated. A quite positive evaluation, without much surprise, given its operating principles and capacity to return ‘depression’ class.

However, one may wonder about the scope and relevance of the exercise presented in this article, given the empirical nature of the method, the preponderance of the choice of parameters used, and the centrality of the visual evaluation. Further, the article and the method suffer from a disappointing shortcoming: the absence of any field confrontation, or real reconnaissance or verification on the ground.

Authors: We added the field survey for the verification on the ground: see sections 1. Introduction; 2.2 Methodological approach, 5. Conclusions, and Figures 9 and 11.

Therefore, reaching the famous but over-used conclusion of "further analysis and methods are needed to be deepened" (l.372) also suggests that this article is perhaps incomplete and premature.

Authors: We clarified this point. For further analyses and methods, we referred to the use of LiDAR on drone (see Conclusions) to improve the resolution of image acquisition.

Such field checking – or at least a test zone - doesn’t seems to have involved such a paramount task as the features at stakes only « cover a total area of 175m2 »… In particular, when, in the course of a sentence (l.352), one learns that the method does not allow to distinguish the sought-after features (looting) from natural depressions which could also exist?

Authors: The field survey enabled us to verify that the features imaged by LiDAR (and derived models) and extracted by Geomorphon are related to archaeological disturbance by grave robbers. In particular, the ground check put in evidence that most of looting pits are practiced with a localized intervention close the opening to the tomb chamber, immediately after the Entrance passage of the tomb. (see Conclusions)

Additional remarks:

Somehow, introduction is an exercise where authors show their knowledge of the state of the art and mobilize references, inflating the overall relevance of their contribution. Here, they are plentiful: more than 10 references just to mention global archaeological looting, as many to talk about Lidar, VTs, automated methods etc. One gets 37 references in less than 80 short lines. But not a single one dealing with lidar use of ancient diggings or excavations. See for instance https://journals.openedition.org/echogeo/14791

Authors: we added the reference suggested by the reviewer

Basically, since the paper deals with tangible features in specific spaces, it is worrisome that the maps and figures almost never have a scale! (figure 3). Furthermore, consistency in the colour codes used between figures would be preferable (e.g., Figures 8 and 9).

Authors: we added the scale.

One notes in passing the absence of a legend for the blue dots in figure 9(A4)

Authors: There was an error in the visualization of the light blue(or cyano) dots instead of red, due to the fact that before exporting the image we forgot to unselect some red dots . The ArcMap selection colors the outline color in cyan and this is maintained also in the exported map

Reviewer 2 Report

In the first part of the introduction, where reference is made to how remote sensing tools have helped to monitor looting, it is recommended to refer to a site where it has been applied. It would serve as a basis for the objectives set out in the article. The methodological approach of the work is correct, since it raises a specific problem around the looting of archaeological sites and shows an alternative for the detection of these looting through the use of remote sensing tools.
However, what is the real usefulness of this tool? One should consider answering the following question: how does this method benefit archeology or how can it be applied to stop the plundering of archaeological sites? If we cannot answer these questions, the method is meaningless, because without utility its use will never become generalized. For this reason, it would be positive if the authors valued the advantages that the use of the proposed method may have after its application to heritage. Whether or not it is a tool that can help stop the destruction of deposits

Or if it simply serves to detect them. In this case, to detect pits.

. Likewise, it would be interesting if the conclusions made reference to the field work necessary to verify, on the ground, the validity of the proposed method. Thus, since looting pits do not have specific characteristics that differentiate them from other types of natural pits or pits made by animals, after checking with the LiDAR method, a field check must be carried out to ensure that looting. In this sense, it may be interesting if the field work that has been carried out at the San Giovenale site to verify the existence of the pits and verify that they are looting pits is exposed. Only in this way will it be possible to value the time and work that the use of this methodology requires.  

Author Response

We thank the reviewer for the valuable comments which enabled us to improve the paper

In the first part of the introduction, where reference is made to how remote sensing tools have helped to monitor looting, it is recommended to refer to a site where it has been applied. It would serve as a basis for the objectives set out in the article. The methodological approach of the work is correct, since it raises a specific problem around the looting of archaeological sites and shows an alternative for the detection of these looting through the use of remote sensing tools.
However, what is the real usefulness of this tool? One should consider answering the following question: how does this method benefit archaeology or how can it be applied to stop the plundering of archaeological sites? If we cannot answer these questions, the method is meaningless, because without utility its use will never become generalized. For this reason, it would be positive if the authors valued the advantages that the use of the proposed method may have after its application to heritage. Whether or not it is a tool that can help stop the destruction of deposits. Or if it simply serves to detect them. In this case, to detect pits.

Authors: we thank the reviewer who allows us to clarify this important point includes in the Conclusion sections herein reported:

For bare areas, optical remote sensing and SAR offer suitable tools for the detection of looting, whereas in wooded areas LiDAR is the unique technology that can provide detailed information under canopy, including archaeological disturbance. The results herein obtained put in evidence the high potentiality of LiDAR to detect and map looted area at single ‘pit’, even if completely covered by trees as evident from field survey. Given the considerable extension of wooded areas in Italy of potential archaeological interest, it is necessary to identify them with appropriate tools such as the LiDAR to understand if they have been disturbed by grave robbers in order to strengthen on-site surveillance and safeguard measures.”

. Likewise, it would be interesting if the conclusions made reference to the field work necessary to verify, on the ground, the validity of the proposed method. Thus, since looting pits do not have specific characteristics that differentiate them from other types of natural pits or pits made by animals, after checking with the LiDAR method, a field check must be carried out to ensure that looting. In this sense, it may be interesting if the field work that has been carried out at the San Giovenale site to verify the existence of the pits and verify that they are looting pits is exposed. Only in this way will it be possible to value the time and work that the use of this methodology requires.  

Authors: In Conclusions we made reference to the field work which allowed us to verify that the features found by the LiDAR data and extracted using Geomorphon actually referred to looting holes, most of them are practiced with a localized intervention close the opening to the tomb chamber, immediately after the dromos (Entrance passage). See also fig. 11.

Reviewer 3 Report

The contribution provides an original approach in the application of remote sensing based on Lidar that complements other proposals for the motorization of archaeological sites. It is not particularly innovative with respect to methodology, techniques or processes applied; however, the model enhancement processes are detailed and well organized. Limitations of the method are also well identified. I think it is a contribution of interest that can be applied to a wide range of cases. It would be interesting to consider in this type of study the different settlement patterns shown by historical societies, which can geographically affect the method applied and the results obtained.

Author Response

We thank the reviewer for the valuable comments which enabled us to improve the paper

Round 2

Reviewer 1 Report

Edits and inclusion of field data references and pictures have been most welcome.